# Factors Influencing Gallstone Formation: A Review of the Literature

**DOI:** 10.3390/biom12040550

**Published:** 2022-04-06

**Authors:** Hao Sun, Jonathan Warren, James Yip, Yu Ji, Shaolong Hao, Wei Han, Yuchuan Ding

**Affiliations:** 1Central Laboratories, Beijing Luhe Clinical Institute, Capital Medical University, Beijing 101199, China; m18810253032@163.com (H.S.); doc.jiyu@foxmail.com (Y.J.); 2Department of Neurosurgery, Wayne State University School of Medicine, Detroit, MI 48201, USA; jonathan.warren2@med.wayne.edu; 3Department of Pathology, John A. Burns School of Medicine, University of Hawaii, Honolulu, HI 96831, USA; jyip@med.wayne.edu; 4Department of General Surgery, Beijing Luhe Clinical Institute, Capital Medical University, Beijing 101199, China; haoshaolong2014@163.com; 5John D. Dingell VA Medical Center, 4646 John R Street, Detroit, MI 48201, USA

**Keywords:** gallstone disease, bile acids, obesity, diabetes, non-alcoholic fatty liver, cardiovascular disease

## Abstract

Gallstone disease is a common pathology of the digestive system with nearly a 10–20% incidence rate among adults. The mainstay of treatment is cholecystectomy, which is commonly associated with physical pain and may also seriously affect a patient’s quality of life. Clinical research suggests that cholelithiasis is closely related to the age, gender, body mass index, and other basic physical characteristics of patients. Clinical research further suggests that the occurrence of cholelithiasis is related to obesity, diabetes, non-alcoholic fatty liver, and other diseases. For this reason, we reviewed the following: genetic factors; excessive liver cholesterol secretion (causing cholesterol supersaturation in gallbladder bile); accelerated growth of cholesterol crystals and solid cholesterol crystals; gallbladder motility impairment; and cardiovascular factors. Herein, we summarize and analyze the causes and mechanisms of cholelithiasis, discuss its correlation with the pathogenesis of related diseases, and discuss possible mechanisms.

## 1. Introduction

Gallstone disease is a common pathology of the digestive system. Nearly 75% of patients with gallstones have no obvious symptoms in the initial stages. As the gallstones progress in development, they may trigger symptoms such as nausea, epigastric colic, diarrhea, anorexia, etc. Eventually, gallstone obstruction can lead to life-threatening conditions such as acute cholangitis, acute cholecystitis, and biliary pancreatitis [1]. At this point, cholecystectomy is necessary. However, this treatment has complications that can seriously threaten a patient’s health and their overall quality of life [2,3]. In 2010, five factors were proposed to promote the formation of gallstones. These include genetic factors; excessive liver cholesterol secretion (resulting in cholesterol supersaturation in gallbladder bile); factors that accelerate the growth of cholesterol crystals and solid cholesterol crystals; impairment of gallbladder motility; and intestinal factors [4]. Additionally, the formation of gallstones is related to factors such as age, gender, weight, and bacterial infection [5,6,7,8]. This review describes and summarizes the causes, formation process and influencing factors of gallstones, and discusses potential mechanisms for their formation.

## 2. Classification and Formation of Gallstones

Gallstones are categorized into several types, including cholesterol, pigment, and mixed stones [9]. Evidence suggests that 90% of cholelithiasis patients have cholesterol stones [10]. Cholesterol sources in the human body mainly consist of ab initio synthesis of acetyl coenzyme, enterohepatic circulation, and food intake. However, because human and animal tissues do not possess enzymes that can degrade the ring structure of this sterol, cholesterol cannot be metabolized to CO_2_ and water in the body. Therefore, to prevent a potentially hazardous accumulation of cholesterol in the body, excess cholesterol must be metabolized into other compounds and/or excreted in the feces.

Cholesterol can be secreted into bile by the liver, and is then carried by bile acid micelles and lecithin, which increase the cholesterol content in bile. Excess cholesterol is carried by cholesterol-rich lecithin cholesterol vesicles which have characteristics of affinity aggregation. They fuse with each other to form large vesicles, resulting in high local cholesterol concentration, and finally form cholesterol hydrate crystals. These cholesterol hydrate crystals then form the initial aggregation nucleus of cholesterol stones, which is an initial and necessary step in the formation of cholesterol stones. In addition, there exists a transitional form for carrying excess cholesterol, phospholipid lamella (disk-like particles), which also play a role in the nucleation of cholesterol crystals [11]. When the cholesterol crystals appear, concentrated granulocytes are recruited, and neutrophils extrude their DNA. This extruded DNA adheres to different cholesterol crystals and, over time, the crystals with neutrophil DNA wrapped around them will coalesce with nearby DNA-wrapped crystals. This pulls the individual DNA-wrapped crystals together, eventually forming larger stones [12]. Clinical studies have found that cholesterol-supersaturated bile is an essential prerequisite for the precipitation of solid cholesterol monohydrate crystals and the formation of cholesterol gallstones.

## 3. Cholesterol and Bile Acid Circulation

### 3.1. Cholesterol Circulation

Cholesterol in the liver can travel through the biliary tract to the small intestine, where some cholesterol is reabsorbed and enters the bloodstream through the lymphatic system, eventually returning to the liver. There are two main sources of cholesterol in the body: biosynthesis and absorption. Cholesterol biosynthesis mainly occurs in the liver. This process converts acetyl-CoA into cholesterol molecules through enzymatic reactions. Endoplasmic reticulum transmembrane protein 3-Hydroxy-3-Methylglutaryl-CoA Reductase (HMGCR) and squalene monooxygenase are rate-limiting enzymes in this process [13]. Part of the cholesterol in hepatocytes is converted into bile salts, and the other part of free cholesterol is pumped out by hepatocytes through ABCG5/8 into the biliary tract. Here, phospholipids form microclumps that are excreted into the intestine through bile secretion [14].

Biosynthetic and dietary cholesterol are absorbed by Niemann–Pick C1-Like Protein1 (NPC1L1) on the membranes of intestinal epithelium and further esterified by ACAT (the alternative name is acyl-coenzymeA: cholesterol acyltransferase), which then enters the bloodstream through the lymphatic system and is eventually absorbed by the liver as chylomicrons [15,16]. The proportion of total cholesterol from diet also depends mainly on the efficiency of absorption of cholesterol from the intestinal tract and the amount of cholesterol consumed daily. Cholesterol absorbed in the small intestine may regulate cholesterol synthesis in the liver through a negative regulatory mechanism, depending on daily food intake.

### 3.2. Bile Acid Cycle

There is an equilibrium between cholesterol and bile acid in bile. The liver is the only organ that synthesizes bile acids (Figure 1). Bile acids are synthesized by cholesterol in liver parenchymal cells, including the classical pathway mediated by cholesterol 7α -hydroxylase (CYP7A1) and the non-classical pathway mediated by sterol 27-hydroxylase (CYP27A1) [17]. The non-classical pathway mainly produces chenodeoxycholic acid (CDCA), while the classical pathway produces CDCA and cholic acid (CA). The newly synthesized binding bile acids are secreted into the capillary bile duct of hepatocytes by the bile salt export pump (BESP (ABCB11)), eventually making their way into the intestine [18].

Binding bile acids can be effectively reabsorbed into intestinal epithelial cells at the terminal ileum by the apical sodium-dependent bile acid transporter (ASBT), which binds to the ileal bile acid binding protein for transport to the basement membrane. In the presence of heterodimer organic solute transporter α/β (OSTα/β) at the terminal cavity of the basement membrane, bile acids are reabsorbed and transported to the liver via the portal vein. At the liver, bile acids are again absorbed from the sinusoidal space of hepatocytes by Na+/taurocholic acid cotransport polypeptide (NTCP (SLC10A1)) [19,20]. The reabsorbed bile acids are then secreted into the biliary system by BESP (ABCB11) in the capillary bile duct of hepatocytes, together with the newly synthesized binding bile acids. Clinically, NTCP is primarily responsible for the uptake of conjugated bile acids (>80%), while members of the Organic Anion Transporting Polypeptide (OATP) family are primarily responsible for the transport of unconjugated or sulphated bile acids to the liver [21].

In addition, free bile acids are passively reabsorbed into intestinal epithelial cells in the small intestine and colon. From there, they are absorbed through the sinusoidal membrane of hepatocytes by OATP and finally processed and secreted into the capillary bile ducts [22]. Bile acids enter the intestinal lumen through the biliary tract, completing the bile acid enterohepatic circulation. Bile acids that are not reabsorbed by liver cells spread throughout the body and may eventually be excreted through the kidneys.

## 4. Factors Influencing Gallstone Formation

Dysfunction of the gallbladder or other parts of the bile-secretion pathway can result in gallstone formation. Given that the bile-secretion pathway is a complex process, there are many reasons for the formation of gallstones. Evidence suggests that gallstones are related to age, gender, female physiological status, obesity, cardiovascular disease, microbiome, sugar metabolism, and various environmental exposures (Figure 2). Based on a large number of mouse and human studies, the interaction of five main factors were proposed. The pathogenesis of cholesterol gallstone disease is precipitated by: genetic factors; excessive cholesterol secretion by the liver (leading to supersaturation of cholesterol in gallbladder bile); rapid phase change by accelerating the growth of cholesterol crystals and solid cholesterol crystals; impairment of gallbladder motility; and intestinal factors. Intestinal factors can be further broken down into two categories: increased cholesterol absorption from the small intestine to the liver, eventually resulting in increased bile secretion, and microbiota that inhabit the intestinal tract. These factors will increase the production or growth of cholesterol crystals, eventually leading to the formation of stones [4,5,6,7,23,24].

### 4.1. Genetic Mechanism of Gallstone Formation

Gallstone-susceptible mice (C57L/J) and gallstone-resistant mice (AKR/J) have been used to better elucidate the genetic components of gallstone formation. Using these mice, it has been described that lithogenic genes 1 and 2 (*Lith1* and *Lith2*) may play a role in gallstone formation. *Lith1*, located on mouse chromosome 2, plays a major role in the determination of liver cholesterol hypersecretion. *Lith2*, located on mouse chromosome 19, regulates the bile salt-dependent flow of bile [25]. The functional counterparts of Mouse *Lith1* and *Lith 2* are ABCG5 and ABCG8 in the human equivalents. ABCG5 and ABCG8 are ATP-binding cassette (ABC) transporters with significant expression in hepatocytes and intestinal cells [26]. These two proteins form heterodimers in the endoplasmic reticulum and are subsequently transported to the apical membrane. In hepatocytes, they transport neutral sterols to bile or they promote active efflux of cholesterol from the enterocyte back into the intestinal lumen for fecal excretion [27]. Inactivation of ABCG5/G8 will result in significantly reduced cholesterol secretion in bile, making the level of cholesterol in liver and plasma very sensitive to changes in dietary cholesterol content. Because of this, hypercholesterolemia, phytosterolemia, and premature coronary heart disease may result [28,29]. However, the overexpression of ABCG5/G8 protein increases cholesterol content in the gallbladder, thus increasing the likelihood of cholesterol crystal precipitation [30]. Subsequently, ABCG5/G8 was found to be associated with cholesterol gallstone disease in patients, and two gallstone associated variants in ABCG5/G8 (ABCG5-R50C and ABCG8-D19H) were identified in Germans, Chileans, Chinese, and Indians. Taking this information into account, these may be the primary promoter genes of gallstones.

### 4.2. Gallbladder Contraction

After ingesting a large amount of food containing fat and protein, the neuroenteropeptide hormone cholecystokinin (CCK), released by endocrine cells of the duodenum, reaches the gallbladder and directly binds with the CCK1 receptor (CCK-1R) on the smooth muscle cells of the gallbladder wall. This triggers contraction of the gallbladder and discharges the concentrated bile into the intestine. CCK-1R is also located in the sphincter of Oddi, pancreas, small intestine, gastric mucosa, and pyloric sphincter. It is responsible for CCK regulation of pancreatic secretion, small intestine transport, gastric emptying, and other digestive processes. Observing CCK or CCK-1R gene knockout mice shows that gallbladder emptying and bile cholesterol metabolism are inhibited, intestinal absorption of cholesterol is increased, and cholesterol stone formation is significantly increased [31]. This suggests that CCK can regulate gallbladder and small intestine motility through the CCK-1R signal cascade, promote small intestine transport, and regulate intestinal cholesterol absorption. This also explains why the abnormal gallbladder motility caused by exogenous cholecystokinin is mainly found in patients with cholesterol stones [32,33].

Clinical studies have found that glucagon-like peptide 1 (GLP-1) receptor agonists have achieved good results in the treatment of type II diabetes, obesity, and other diseases. However, such drugs have a negative impact on the gallbladder and seem to increase the risk of gallbladder-related diseases [34]. In one study after acute injection of the glp-1r agonist exendin-4, there was no significant change in gallbladder volume in mice. When combined with CCK injection, exendin-4 reduced the emptying ability of the gallbladder. The effect of the glp-1r agonist for 12 weeks on patients with type 2 diabetes mellitus was not significant. In addition, the mRNA transcription level of GLP-1R in the gallbladder of mice was low, suggesting that GLP-1 has a more indirect effect on the gallbladder. This further suggests that GLP-1 may be related to slowing down of upper gastrointestinal motility. However, the molecular mechanism is still unclear [35].

### 4.3. Microbiome

The various flora in the body are in a dynamic balance and, when disturbed, many tissues and organs are affected. This complex system of microorganisms also exists in bile, and the occurrence of gallstones is closely related to abnormalities with flora. In almost all stages of bile formation, the microbiota of the gastrointestinal and biliary tracts are involved, including the regulation of lipid metabolism, cholesterol metabolism, biotransformation, and enterohepatic circulation of bile acids [24].

Microbiome in the biliary tract. Studies have shown the presence of living bacteria in gallstones. The flora in the biliary tract and duodenum are highly homologous and closely related to the formation of gallstones. Microorganisms can enter the biliary system from the duodenum by migrating through the sphincter of Oddi. They can also spread hematogenously to the liver and from there into bile [36,37]. When in bile, microorganisms play an important role as nucleating factors, leading to the formation of pigment and cholesterol gallstones [38].

The properties of bacteria that reside in the gallbladder can control the formation of gallstones. Bacteria that produce beta-glucuronidase and phospholipase in the bile yield a higher percentage of pigment in the stones, while bacteria that cause mucus abnormalities are more likely to lead to the formation of cholesterol stones [39,40]. Biofilm-forming bacteria in the gallbladder, bile, and gallstones are closely associated with gallstone formation [41]. For example, biofilms are formed during the formation of pigment stones, and the aggregating factor in this case is the glycocalyx (anionic glycoprotein) [42]. Differences in the functional metagenomes of microbial communities have been found by comparing pigment gallstones and cholesterol gallstones. Gram-positive bacteria were predominant in most of the cholesterol gallstones examined, whereas they were not found in the pigment stones. A high proportion of genes involved in carbohydrate metabolism were found in the pigment stones, whereas genes dominating protein metabolism were more active in the cholesterol stones. *Helicobacter pylori* is a Gram-negative, spiral-shaped, motile microorganism [43]. The presence of *H. pylori* in patients with symptomatic gallstone disease (GSD) has been shown to promote the formation of gallstones. However, this finding is still controversial and more data are required for adequate discussion of this topic [44].

Oral flora. Microflora of the oral cavity affects the secretion of cholecystokinin [45], the main factor involved in the emptying and filling of the gallbladder [46]. A microbiome changes the expression of mucin genes (MUC1, MUC3, and MUC4 genes) through immunomodulation, thereby changing the accumulation of mucin gel, which is the nucleation matrix for the formation of cholesterol gallstones in the gallbladder [46]. Studies have shown that *H. pylori* and enterohepatic strains of Helicobacter contribute to the formation of cholesterol gallstones [47,48,49].

The composition of the gut and biliary tract microbiome varies significantly in patients with GSD and in healthy subjects [50]. In patients with GSD, microbial diversity is reduced, beneficial bacteria such as *Roseburia* are reduced, and an overgrowth of bacteria of the Proteobacteria type—including a wide range of pathogenic microorganisms such as *Escherichia*, *Salmonella*, *Vibrio* and *Helicobacter*—more easily occurs [51]. Disorders of bile acid metabolism are the leading factors in the pathogenesis of cholesterol GSD [25].

Gut microbes. Gut microbiota-mediated biotransformation of the bile acid pool regulates bile acid signaling by influencing the activation of host bile acid receptors, such as the nuclear receptor farnesoid X receptor (FXR). The role of FXR in liver cells and intestinal cells is recognized as a regulator of bile acid, lipid, and glucose balance [52,53]. In the intestine, bile acids directly bind to and activate the bile acid receptors FXR and fibroblast growth factor 15/19 (FGF15/19) gene expression. FGF15/19 inhibits the synthesis of bile acids by reducing the expression of CYP7A1, which plays a negative feedback role in bile acid synthesis [54]. In fact, disturbances in the intestinal microbiota and changes in the composition of bile can adversely affect the metabolism of bile acids and the balance of glucose and cholesterol, leading to the development of gallstones [55,56].

Gut microbes lower cholesterol in bile. Bifidobacteria have been proven to lower cholesterol in bile by assimilation or precipitation [39,57]. A meta-analysis showed that probiotics (*L. acidophilus*, *B. lactis*, *VSL#3*, and the *L. plantarum* group) can significantly reduce total serum cholesterol [58]. The consumption of a BSH-positive strain of Lactobacillus significantly reduced cholesterol in patients with hypercholesterolemia [59].

### 4.4. Effect of Estrogen in Gallstone Formation

According to grouping analysis of cholelithiasis, the number of female patients of all ages with gallstones is significantly higher than that of men. The incidence rate of gallbladder diseases in women is further increased during pregnancy, which has become the second most common indication of non-obstetric intervention during pregnancy [60,61,62]. Furthermore, the incidence rate of gallbladder disease in women who have had multiple pregnancies is higher than that of those who have been pregnant once [63]. The importance of estrogen in terms of cholelithiasis is well documented. Estrogen, such as 17β-estradiol (E2), is a major female steroid hormone which plays an important role in health and disease [30]. As a steroid, estrogen has liposoluble properties which allows it to passively diffuse into cells and play the role of a transcription factor. After entering cells, it directly binds to ESR1 and ESR2 receptors and initiates changes in receptor tertiary and quaternary structures. As a result, active complexes that regulate transcription are formed [64]. When E2 reaches the liver, it also passively diffuses into cells and increases liver secretion of cholesterol into bile, thus increasing the cholesterol saturation in bile and the risk of cholelithiasis. It has been confirmed that ESR1, rather than ESR2, plays a more major role in the formation of cholesterol gallstones in mice induced by high dose E2. E2 has also been shown to play important roles in health and disease [65,66]. It regulates a wide range of biological processes, including reproduction, cardiovascular function, hepatobiliary secretion, metabolic processes, nerve function, and inflammation. There is a large amount of clinical evidence suggesting that oral contraceptive steroids and conjugated estrogen play a significant part in promoting cholesterol stone formation in premenopausal women [67,68,69,70]. The classical estrogen regulatory pathway involves E2 promotion of cholesterol biosynthesis and liver secretion of bile cholesterol through the “e2-esr1-srebp-2” pathway. During estrogen treatment or in times of increased blood estrogen concentration, synthesis of cholesterol increases mainly by estrogen-induced stimulation of sterol regulatory element binding protein-2 (SREBP-2) [71]. These changes lead to excessive secretion of newly synthesized cholesterol, supersaturation of bile, and easily lead to cholesterol precipitation and gallstone formation. Estrogen-activated ESR1 also stimulates the activity of ABCG5 and ABCG8, which are only expressed in hepatocytes and intestinal cells. These two proteins form heterodimers in the endoplasmic reticulum and are then transported to the apical membrane. There, they transport neutral sterols to bile or the intestinal lumen which promotes the secretion of bile cholesterol, eventually leading to the supersaturation of cholesterol in bile [29].

G protein-coupled receptor 30 (GPR30), a newly discovered estrogen receptor in humans, is produced by the gallstone gene *lith18* [4,72,73]. It was found that E2 can effectively bind and activate GPR30 and ER-α [74]. In order to distinguish the role of Er-α and GPR30 in stone formation, a mouse model of ovariectomized female wild type, GPR30 gene knockout, ER-α gene knockout, and GPR30/ER-α double gene knockout was constructed. It was found that E2 activated GPR30 and ER-α produced liquid crystal and amorphous metastable intermediates. These evolved into cholesterol monohydrate crystals from supersaturated bile. In addition, the cholesterol crystal of GPR30/ER-α double knockout mice decreased significantly. This suggests that GPR30 and ER-α have a synergistic effect on the formation of gallstones induced by E2 [73,75]. Because GPR30 is mainly located in the endoplasmic reticulum rather than the nucleus of hepatocytes, E2 may activate GPR30 through the signal cascade of epidermal growth factor receptor, thus inhibiting the classical pathway of hepatic cholesterol 7α-hydroxylase and bile acid synthesis. This results in excessive cholesterol production, leading to increased cholesterol secretion by the liver and increased likelihood of bile stone formation [75].

### 4.5. Obesity and Gallstone

Nonalcoholic fatty liver disease (NAFLD) is an important risk factor for gallstone formation. The abnormally low expression of aquaporin 8 (AQP8) mediated by hypoxia inducible factor-1α (HIF-1α) in NAFLD seems to explain this situation [76]. HIF-1α is an important transcription factor regulating gene expression of oxygen transfer, cell growth, and redox homeostasis that promotes an adaptive response to hypoxic conditions resulting in greater cell survival [76,77]. Given this, it makes sense that in the liver HIF-1α mainly exists in the area around the hepatic vein. During the development of hepatic steatosis, lipid accumulation significantly increases the size of hepatocytes, thereby reducing hepatic sinusoidal perfusion and microcirculation, and ultimately leading to liver hypoxia [78]. In one study examining the upregulated expression of AQP8, a water channel protein responsible for the secretion of liver water into the bile duct [79,80,81], researchers found a significant 35% increase in bile flow, diluted bile lipid concentration in gallbladder and hepatobiliary juice by 36%, and alleviation of gallbladder inflammation. As a result, cholesterol crystal formation was inhibited in the liver-specific HIF-1α knockout mice. On the contrary, activation of the HIF-1α pathway in diet-induced fatty liver has been shown to accelerate the formation of gallstones in wild-type mice. In addition, the increased expression of HIF-1α and its downstream targets in the liver suggests that HIF-1α may play an important role in the formation of cholesterol gallstones in patients with NAFLD [2].

### 4.6. Carbohydrate Metabolism

The classic role of bile in the digestion and absorption of fat is well documented. In addition, the gallbladder also plays a physiological role in glucose, fat, and energy homeostasis. Both GSD and cholecystectomy can reduce insulin sensitivity [82], which suggests that obesity is not a correlate between gallstone and insulin resistance (IR) but is a common risk factor for both. GSD and cholecystectomy increase triglyceride content in the liver, and possibly increase IR in the liver as well. On the other hand, the gallbladder not only regulates the secretion and transport of bile acids, but also affects the homeostasis of lipids and glucose, which may affect whole body energy consumption [83,84]. In addition, experimental evidence suggests that liver IR may promote GSD by increasing the diagenesis of bile.

## 5. Gallstones, Cardiovascular and Cerebrovascular Diseases

Recent studies have found that GSD is also closely related to the occurrence of cardiovascular disease (CVD), and the presence of GSD increases the incidence of CVD [85,86]. According to a meta-analysis of 10 published study cohorts, Zhao et al., found that patients with GSD were at higher risk of hypertension, diabetes, coronary heart disease, atrial fibrillation, and hyperlipidemia. Additionally, they found that GSD was associated with a 1.23-fold increase in the incidence of cardiovascular and cerebrovascular diseases [87]. In a cohort of 5,928 subjects established by Daniel et al., gallstone disease was associated with all cardiovascular disease (hazard ratio (HR) 1.36, 95% confidence interval (CI) (1.17; 1.59)) and to the subgroups’ coronary artery (HR1.34, 95% CI (0.10; 1.64)), cerebrovascular (HR 1.22, 95% CI (0.97; 1.52)), and peripheral artery disease (HR 1.57,95% CI (1.15; 2.13)) [88]. Gallstones and cardiovascular disease share common risk factors such as age, sex, obesity, and disorders of lipid and glucose metabolism, all of which are major risk factors for metabolic syndrome. Metabolic syndrome is strongly associated with coronary artery disease and gallstones may be considered a biliary feature of this syndrome [89,90].

It is well-known that cholesterol is mainly carried by lipoproteins in plasma and by micelles and vesicles in bile. If excess cholesterol is accumulated in the arterial wall, it leads to atherosclerosis and causes cardiovascular disease. If excess cholesterol cannot be dissolved in bile by the bile salts and/or phospholipids, it precipitates as plate-like solid cholesterol monohydrate crystals, thus leading to the formation of cholesterol gallstones in the gallbladder and/or the bile duct [27].

Cholesterol is a major component of most gallstones and is a major component of atherosclerotic plaques. Cholesterol accumulation is a major cause of atherosclerotic CVD and GSD [91]. In the cardiovascular system, such metabolic abnormalities usually lead to the accumulation of excessive cholesterol esters in the arterial wall, leading to clinical atherosclerosis, which mainly occurs in the heart and brain, leading to cardiovascular and cerebrovascular diseases [88]. Cholesterol is present in the blood in the form of lipoproteins and is involved in the transport of lipids. High-density lipoprotein (HDL) is a protective factor of CVD, which can transport cholesterol in surrounding tissues and convert it into bile acid or be discharged directly from the intestinal tract through bile into the biliary system. It has a significant negative correlation with the degree of arterial lumen stenosis. Low density lipoprotein (LDL) carries cholesterol into peripheral tissue cells. When LDL is excessive, the carried cholesterol accumulates in the arterial wall, increasing susceptibility to arteriosclerosis, cardiovascular, and cerebrovascular diseases [92].

As mentioned above, ABG5/8 can transport cholesterol to the intestine and bile. Abnormally elevated expression of ABG5/8 can reduce cholesterol concentration in blood and inhibit the occurrence of atherosclerosis, but it can increase cholesterol concentration in bile and increase the risk of cholesterol-type stones. NPC1L1 mediates cholesterol uptake from the gut to intestinal cells and liver reuptake of cholesterol from bile to hepatocytes, thus counteracting ABG5/8 function. NPC1L1 is a transmembrane protein highly expressed in the intestinal tract, especially in the lumen membrane of mammalian intestinal epithelial cells, which mediates intestinal cholesterol absorption [93,94,95]. NPC1L1 is a target molecule of ezetimibe, a cholesterol absorption inhibitor clinically used in the treatment of dyslipidemia [96]. Inhibition of intestinal activity of NPC1L1 or genetic variants reduces cholesterol uptake, lowers plasma LDL cholesterol concentrations, and prevents atherosclerosis and ischemic vascular diseases. Inhibition of hepatic NPC1L1 may lead to an increase in bile cholesterol concentration, thereby promoting the formation of cholesterol gallstones [97].

Abnormal inflammation is involved in the development of GSD and CVD. Reducing the expression of certain inflammatory factors has been shown to improve CVD [98]. Some inflammatory factors, including von Willebrand factor (vWF), lectin-like oxidized low-density-lipoprotein receptor-1 (LOX-1), as well as soluble urokinase plasminogen activator receptor (suPAR), have been proposed to be associated with CVD [99,100,101,102]. Inflammatory processes in GSD may promote atherosclerosis, vascular lesions in the cerebrovascular system, and increase the risk of cardiovascular disease. Related liver diseases, including nonalcoholic fatty liver disease and suppurative liver abscesses, have been identified as risk factors for subsequent cardiovascular disease [103].

## 6. Conclusions

New evidence continues to uncover the many roles the gallbladder plays in different physiologic mechanisms and its importance in broader physiological function. The relationship between cholelithiasis and disease processes such as diabetes mellitus, nonalcoholic fatty liver disease, obesity, and insulin resistance suggests that the gallbladder is an important link in the multi-directional communication between different tissues mediated by bile acids, gut hormones, hepatocyte factors, and adipokines. The gallbladder serves as a link in coordinating metabolic homeostasis, maintenance of normal body composition, and insulin sensitivity. We summarized the important feedback mechanisms of FXR and GLP-1 in cholesterol metabolism and insulin sensitivity, discussed the possible important role of GLP-1 in gallbladder contraction disorders, and examined the relationship between cholelithiasis and metabolism. With further understanding of the processes involved in gallstone formation, one may be able to elucidate its role for better treatment of gallbladder disease in the future.

## Figures and Tables

**Figure 1 biomolecules-12-00550-f001:**
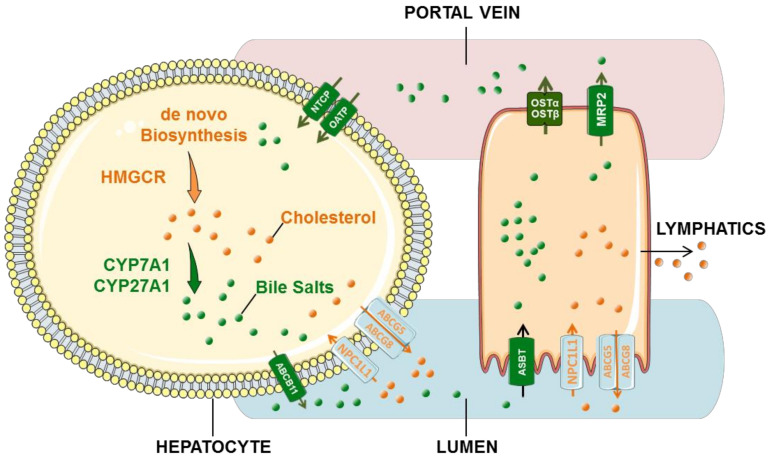
Cholesterol synthesis metabolism and bile acid hepatoenteric circulation. Cholesterol can be de novo synthesized. Part of the cholesterol in hepatocytes is converted into bile salts, and the other part of free cholesterol is pumped out by hepatocytes through ABCG5/8 into the biliary tract. Cholesterol is absorbed by NPC1L1 on the membranes of intestinal epithelium, which then enters the bloodstream through the lymphatic system. The newly synthesized binding bile acids are secreted into the capillary bile duct of hepatocytes by ABCB11, eventually making their way into the intestine. Binding bile acids can be effectively reabsorbed into intestinal epithelial cells at the terminal ileum by ASBT, which binds to ileal bile acid binding protein for transport to the basement membrane. In the presence of heterodimer OSTα/β at the terminal cavity of the basement membrane, bile acids are reabsorbed and transported to the liver via the portal vein. Orange represents cholesterol. Dark green represents the bile acid cycle. NPC1L1:Niemann–Pick C1-Like Protein1; ABCG5/8: ATP Binding Cassette Subfamily G Member 5/8; CYP7A1: cholesterol 7α–hydroxylase; CYP27A1: sterol 27-hydroxylase; ABCB11: ATP-Binding Cassette Sub-Family B Member 11; ASBT: the apical sodium-dependent bile acid transporter; OSTα/β: organic solute transporter α/β.

**Figure 2 biomolecules-12-00550-f002:**
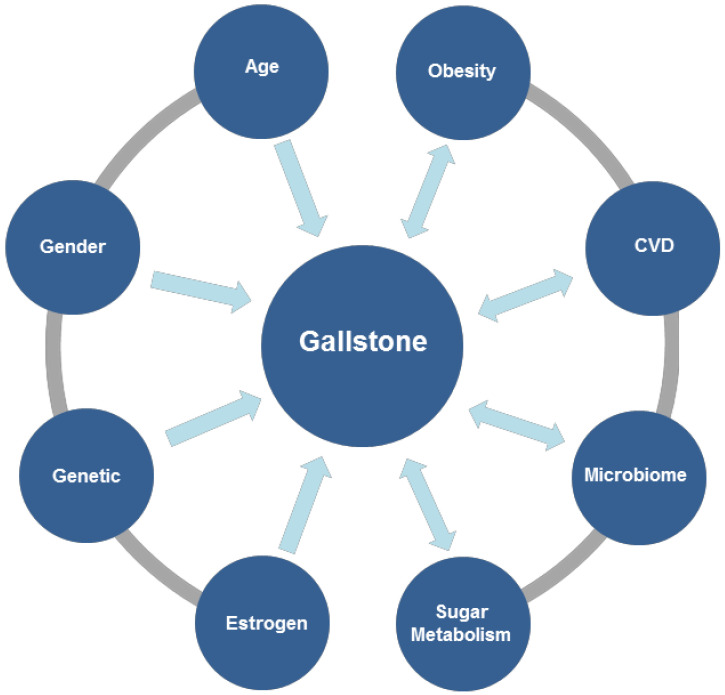
Influence factors of gallstones. CVD: cardiovascular disease.

## Data Availability

No data was reported in this study.

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
