# Peer review of "Factors Influencing Gallstone Formation: A Review of the Literature"

_biomolecules, 2022, doi:10.3390/biom12040550_

Round 1

Reviewer 1 Report

Thank you for the significantly improved article. I have one concern regarding the structure of the manuscript.

Part 4 is entitled "Factors influencing gallstone formation" and further goes 4.6 under the name "Gallstones and cardiovascular and cerebrovascular diseases" I really doubt that cardiovascular diseases increase the formation of gallstones. I think section 4.6 is interesting, but should be somehow separated from part 4.

Author Response

Thank you for the significantly improved article. I have one concern regarding the structure of the manuscript. Part 4 is entitled "Factors influencing gallstone formation" and further goes 4.6 under the name "Gallstones and cardiovascular and cerebrovascular diseases" I really doubt that cardiovascular diseases increase the formation of gallstones. I think section 4.6 is interesting, but should be somehow separated from part 4 Response : Thank you suggestion. The section 4.6 has been separated from part 4 and as part 5.And change part4.7 to part4.6. The content title has been changed as follows. “4.6. Gallstones and cardiovascular and cerebrovascular diseases 4.7. Carbohydrate Metabolism 5. Conclusion” Change to “4.6. Carbohydrate Metabolism 5. Gallstones, cardiovascular and cerebrovascular diseases 6. Conclusion”

Reviewer 2 Report

This is a very interesting review of the literature on gallstone formation. 

minor issues:

Point 1

 Introduction - lines 31-33 -"Eventually, gallstone obstruction can lead to life-threatening conditions such as acute cholangitis, acute cholecystitis, biliary pancreatitis, and cancer of the biliary tract" - chronic inflammation leads to cancer not direct obstruction by stones

Point 2

Introduction - lines 33-34 -

"However, this treatment modality may seriously threaten a patient’s health and their overall quality of life" - I hope you mean treatment complications.

Point 3

Classification and Formation of Gallstones  - lines 45-46- "Gallstones are categorized into several types, including cholesterol, pigment, and phosphate gallstones" - traditional classification consists of cholesterol, pigment, and mixed stones. Other classifications for example https://doi.org/10.1371/journal.pone.0074887

Point 4

Conclusion - conclusion is concentrated on the gallbladder, not on gallstone formation. 

Author Response

This is a very interesting review of the literature on gallstone formation. minor issues: Point 1: Introduction - lines 31-33 –“Eventually, gallstone obstruction can lead to life-threatening conditions such as acute cholangitis, acute cholecystitis, biliary pancreatitis, and cancer of the biliary tract” - chronic inflammation leads to cancer not direct obstruction by stones. Response 1: Thank you for the valuable comments.We've deleted “cancer of the biliary tract”. This change is as follows:” Eventually, gallstone obstruction can directly lead to life-threatening conditions such as acute cholangitis, acute cholecystitis and biliary pancreatitis.” Point 2: Introduction - lines 33-34 -"However, this treatment modality may seriously threaten a patient’s health and their overall quality of life" - I hope you mean treatment complications. Response 2: Thank you suggestion. We've changed the “treatment modality” to “treatment complications”. This change is as follows:”However, this treatment may result in complications that seriously threaten a patient’s health and their overall quality of life” Point 3: Classification and Formation of Gallstones - lines 45-46- "Gallstones are categorized into several types, including cholesterol, pigment, and phosphate gallstones" - traditional classification consists of cholesterol, pigment, and mixed stones. Other classifications for example. Response 3: Thanks for your correction. We have modified it according to your opinion and added that literature. This change is as follows:”Gallstones are categorized into several types, including cholesterol, pigment, and mixed stones.” Point 4: Conclusion - conclusion is concentrated on the gallbladder, not on gallstone formation. Response 4: We have made changes accordingly. We deleted the excessive discussion of gallbladder. This change is as follows:”New evidence continues to uncover the many roles the gallbladder plays in different physiologic mechanisms and its importance in broader physiological functions. The relationship between cholelithiasis and disease processes such as diabetes mellitus, nonalcoholic fatty liver disease, obesity, and insulin resistance suggests that the gallbladder is an important link in the multi-directional communication between different tissues mediated by bile acids, gut hormones, hepatocyte factors, and adipokines. The gallbladder serves as a link in coordinating metabolic homeostasis, maintenance of normal body composition and insulin sensitivity. We summarized the important feedback mechanisms of FXR and GLP-1 in cholesterol metabolism and insulin sensitivity, discussed the possible important role of GLP-1 in gallbladder contraction disorders, and examined the relationship between cholelithiasis and metabolism. With further understanding of the processes involved in gallstone formation, one may be able to elucidate its role for better treatment of gallbladder disease in the future.”

Reviewer 3 Report

Here are my suggestions in order to improve the work:

  • Line 17-18: The sentence makes no sense
  • Line 76: Define HMHCR (it's the first time it appears in the manuscript)
  • Line 82: alternative name is also acyl-coenzymeA:cholesterol acyltransferase
  • In the section of cholesterol circulation: please improve and explain how the synthesized cholesterol reaches circulation
  • Line 91: change number figure, it should be 1
  • Figure 1 caption should be further developed and improved. Briefly explain the process and name full name of enzymes
  • Line 132: figure 2 is misplaced. It should be on line 127 and instead saying many other factors, it should enumerate all the ones included in the figure
  • Line 143: remove first sentence of mouse as an important model. This is vague and its general for all the diseases.
  • Name of bacteria should be in italics throught the manuscript
  • Line 220: Define GSD (it has not been done before in the manuscript)
  • Line 351: cholesterol is present in the form of lipoproteins, not apolipoproteins!!!
  • Line 359: a reference is missing
  • Line 374: meaning of GD

Author Response

Here are my suggestions in order to improve the work: Point 1:Line 17-18: The sentence makes no sense Response 1: Thank you for the comments. We have removed it. Point 2:Line 76: Define HMHCR (it's the first time it appears in the manuscript) Response 2: Thank you suggestion. We have defined 3-Hydroxy-3-Methylglutaryl-CoA Reductase (HMHCR). Point 3: Line 82: alternative name is also acyl-coenzymeA:cholesterol acyltransferase In the section of cholesterol circulation: please improve and explain how the synthesized cholesterol reaches circulation Response 3: Thank you suggestion. We have made the changes accordingly, as follows: ”Cholesterol in the liver can travel through the biliary tract to the small intestine, where some cholesterol is reabsorbed and enters the bloodstream through the lymphatic system ,eventually returning to the liver. There are two main sources of cholesterol in the body: biosynthesis and absorption. Cholesterol biosynthesis mainly occurs in the liver. This process converts acetyl-CoA into cholesterol molecules through enzymatic reactions. Endoplasmic reticulum transmembrane protein 3-Hydroxy-3-Methylglutaryl-CoA Reductase (HMGCR) HMGCR and squalene monooxygenase are rate-limiting enzymes in this process [13]. Part of the cholesterol in hepatocytes is converted into bile salts, and the other part of free cholesterol is pumped out by hepatocytes through ABCG5/8 into the biliary tract. Here, phospholipids form microclumps that are excreted into the intestine through bile secretion [14].” Point 4: Line 91: change number figure, it should be 1; Figure 1 caption should be further developed and improved. Briefly explain the process and name full name of enzymes Response 4: We have corrected the error accordingly. We have written the figure 1 legend as follows. “Figure 1. Cholesterol synthesis, metabolism and bile acid hepatoenteric circulation. Cholesterol in the liver can be de novo synthesized. Part of the cholesterol in hepatocytes is converted into bile salts, and the other part of free cholesterol is pumped out by hepatocytes through ABCG5/8 into the biliary tract. Cholesterol is absorbed by NPC1L1 on the membranes of intestinal epithelium, which then enters the bloodstream through the lymphatic system. Bile acids are synthesized by cholesterol in liver parenchymal cells, including the classical pathway mediated by CYP7A1 and the non-classical pathway mediated by CYP27A1. The newly synthesized binding bile acids are secreted into the capillary bile duct of hepatocytes by ABCB11, eventually making their way into the intestine. Binding bile acids can be effectively reabsorbed into intestinal epithelial cells at the terminal ileum by ASBT, which binds to ileal bile acid binding protein for transport to the basement membrane. In the presence of heterodimer OSTα/β at the terminal cavity of the basement membrane, bile acids are reabsorbed and transported to the liver via the portal vein. Orange represents cholesterol. Dark green represents the bile acid cycle. NPC1L1:Niemann-pick C1-like Protein1, ABCG5/8: ATP Binding Cassette Subfamily G Member 5/8,CYP7A1:cholesterol 7α –hydroxylase,CYP27A1:sterol 27-hydroxylase,ABCB11: ATP-Binding Cassette Sub-Family B Member 11, ASBT: the apical sodium-dependent bile acid transporter, OSTα/β:organic solute transporter α/β .” Point 5: Line 132: figure 2 is misplaced. It should be on line 127 and instead saying many other factors, it should enumerate all the ones included in the figure Response 5: Thank you suggestion. We have made the changes accordingly. This change is as follows:”Evidence suggests that gallstones are related to age, gender, female physiological status, obesity, cardiovascular disease, microbiome, sugar metabolism and various environmental exposures (Figure 2).” Point 6: Line 143: remove first sentence of mouse as an important model. This is vague and its general for all the diseases. Response 6: Thank you for the comment. We have removed it accordingly . Point 7:Name of bacteria should be in italics throught the manuscript Response 7: Thank you suggestion. We have made the changes accordingly. Point 8:Line 220: Define GSD (it has not been done before in the manuscript) Response 8: Thank you suggestion. We have defined GSD (Gallstone disease). Point 9:Line 351: cholesterol is present in the form of lipoproteins, not apolipoproteins!!! Response 9: Sorry for the error . We have corrected it accordingly. Point 10:Line 359: a reference is missing Response 10:Thank you suggestion. We have added reference 92 (PMID:31748177). Point 11:Line 374: meaning of GD Response 11: Sorry for the errors. It should be "GSD".

This manuscript is a resubmission of an earlier submission. The following is a list of the peer review reports and author responses from that submission.

Round 1

Reviewer 1 Report

Thank you for the opportunity to review this article. It seems that the authors did put a lot of effort in going through the literature, however the review lacks novelty. Here are other issues that I have noted:

Introduction: Lines 39-40 sentence should be rewritten “Additionally, the formation of gallstones is related factors such as age, gender, weight, bacterial infection, and other factors” 

Citations in the article are not very accurate. I did not go through them all but i.e. citation 7, line 52 does not speak about gallstone classification at all. Furthermore citations 8 and 9 in line 58 are also not accurate for the sentence. Citation numbering is out of order, after the reference numbered 11 a refence numbered 23 appears further in the article. These mistakes and inaccuracies lead to distrust of all the other references. 

Furthermore, parts 2 and 3 of the article mainly focuses on describing physiological bile production aspects. In my opinion this does not add anything new to the literature, it should be shortened and focus more on the pathology how the gallstones form, or the parts 2 and 3 could be completely discarded. 

In the part 4.5 of the article the authors do not really explain how carbohydrate metabolism causes gallstone formation, but mainly explains how bile acids effect the carbohydrate metabolism and this is not the scope of the review.

Reviewer 2 Report

It is a very well-organized content regarding the formation of gallstones. However, I ask for some minor corrections.
First, the abbreviations (GS,CVD, etc.) are shown in Fig. 1, but it is necessary to explain the abbreviations.

Lines 174 to 178 are difficult to understand due to the long clauses.
Please make sure you are using spaces appropriately. (for example, in the description of figure2.)
The initial letter of cardiovascular on line 316 can be lowercase.
What is the meaning of coarse on line 318?
'More and more evidence is' is preferably 'More and more evidences are' on line 419.